

# Determination and significance of upper-tropospheric humidity

Klaus Gierens[1] and Kostas Eleftheratos[2]

[1]Deutsches Zentrum für Luft- und Raumfahrt, Institut für Physik der Atmosphäre, Oberpfaffenhofen, Germany

[2]Department of Geology and Geoenvironment, National and Kapodistrian University of Athens, Athens, Greece

*Correspondence to:* Klaus Gierens (klaus.gierens@dlr.de)

**Abstract.** We present a novel retrieval for upper-tropospheric humidity (UTH) from HIRS channel 12 radiances that successfully bridges the wavelength change from $6.7$ to $6.5\,\mu$m that occurred from HIRS 2 on NOAA 14 to HIRS 3 on NOAA 15. The jump in average brightness temperature ($T_{12}$) that this change caused (about $-7\,$K) could be fixed with a statistical intercalibration method (Shi and Bates, 2011). Unfortunately, the retrieval of UTHi based on the intercalibrated data was not satisfying

at the high tail of the distribution of UTHi. Attempts to construct a better intercalibration in the low $T_{12}$ range (equivalent to the high UTHi range) were either not successful (Gierens et al., 2018) or required additional statistically determined corrections to the measured brightness temperatures (Gierens and Eleftheratos, 2017).

     The new method presented here is based on the original one (Soden and Bretherton, 1993; Stephens et al., 1996; Jackson and Bates, 2001), but it extends linearisations in the formulation of water vapour saturation pressure and in the temperature-dependence

of the Planck function to second order. To achieve the second-order formulation we derive the retrieval from the beginning, and we find that the most influential ingredient is the use of different optical constants for the two involved channel wavelengths ($6.7$ and $6.5\,\mu$m). The result of adapting the optical constant is an almost perfect match between UTH data measured by HIRS 2 on NOAA 14 and HIRS 3 on NOAA 15 on 1004 common days of operation. The method is applied to both UTH and UTHi, the upper-tropospheric humidity with respect to ice. For each case retrieval coefficients are derived.

We present a number of test applications, e.g. on computed brightness temperatures based on high-resolution radiosonde profiles, on the brightness temperatures measured by the satellites on the mentioned 1004 common days of operation. Further we present time series of the occurrence frequency of high UTHi cases and we show the overall probability distribution of UTHi. The two latter applications expose clear indications of moistening of the upper troposphere over the last 35 years.

     Finally, we discuss the significance of UTH. We state that UTH algorithms cannot be judged for their correctness or incor-

rectness, since there is no true UTH. Instead, UTH algorithms should fulfill a number of usefulness-postulates, that we suggest and discuss. In the course of this discussion an alternative method to estimate the weighting function is presented.



# 1 Introduction

Upper tropospheric humidity (UTH) is a climate parameter which is important to monitor and study with a view to determining long–term trends. The effective determination of these trends requires high quality and continuous water vapour measurements in the upper troposphere. Tropospheric and stratospheric water vapour have been measured over the past 50 years by a large

number of individuals and institutions using a variety of in situ and remote sensing measurement techniques. Measurement results by different instruments are widely spread in the literature (e.g. Sherwood et al., 2010, and references therein). Instrumentation has evolved from a small number of manually operated in situ instruments to automatic devices deployed on balloon and aircraft platforms, and more recently to high precision sensors on satellites. However, only a limited number of measurements of relative or specific humidity using a single instrument type have records longer than 10 years (Kley et al.,

10  2000).

Satellite sensors have the advantage of providing measurements at near–global scale. For climate variability studies it is important to pay attention to the continuity of these measurements, for the determination of long–term changes both in the stratosphere and the upper troposphere. In this respect, satellite missions are planned to provide overlap with existing instruments in orbit. This allows the comparison between different instruments during common periods of observation, which helps

scientists to perform inter–satellite calibrations and produce continuous long–term data sets. Such a long–term data set comes from the National Oceanic and Atmospheric Administration (NOAA) polar orbiting satellites in which measurements are based on the High–Resolution Infrared Radiation Sounder (HIRS). The first series of the satellites from N06 to N14 carried version 2 of the HIRS instrument (HIRS 2), N15 carried version 3 of the HIRS instrument (HIRS 3), while N16 and N17 carried a newer version of the HIRS instrument, version 4 (HIRS 4). Ongoing satellite missions N19 and M02 (part of European Organisation

for the Exploitation of Meteorological Satellites, EUMETSAT) both operate with HIRS 4 instruments.

The transition from HIRS 2 to HIRS 3 in the late 1990s, accompanied with a change in the central measurement wavelength of channel 12 from $6.7\,\mu$m to $6.5\,\mu$m, brought an unwanted change in the measured brightness temperatures at channel 12 (the water vapour channel; $T_{12}$ for convenience). This technical alteration split the continuous HIRS time series in two parts. The problem of discontinuity in the long–term time series was recently solved by Shi and Bates inter–satellite calibrations

(Shi and Bates, 2011), and indeed results presented by Chung et al. (2016) indicated the success of their inter–calibration method throughout the whole time series. Later on, Gierens and Eleftheratos (2017) noted that the inter–calibration works well for the mean $T_{12}$ but does not account for the low $T_{12}$ values (data found at the low tail of the distribution of $T_{12}$). The authors came up with a second statistical correction procedure that brings HIRS 3 levels down to HIRS 2 levels, which takes effect at cold temperatures when $T_{12} < 235\,$K. The method appeared to correct satisfactorily the low $T_{12}$ values, such that jumps in the

time series from HIRS 2 to HIRS 3 where no longer apparent at the low $T_{12}$ temperature range.

Very recently, Gierens et al. (2018) tested the physics behind the statistical inter–calibration of Shi and Bates (2011), wondering whether it is right from a physical point of view to combine measurements by two instruments which sense different layers of the upper atmosphere. They compared $T_{12}$ data calculated by radiative transfer modelling of large sets of temperature and relative humidity profiles, using the HIRS 2 and HIRS 3 spectral response functions. By applying appropriate corrections



to the modelled $T_{12}$, corrections that take into account variability in mid–tropospheric humidity, they calculated $T_{12}$ data consistent with those found by Shi and Bates' independent inter–calibration method. Nevertheless, the Gierens et al. (2018) physics–based correction method failed to correct the data at the low tail of the $T_{12}$ distribution, indicating the necessity of additional corrections to the low $T_{12}$ values as proposed by Gierens and Eleftheratos (2017).

The study that forms the basis for retrieving UTH is that of Soden and Bretherton (1993), in the following SB93. Their method was based upon observations of clear sky $6.7\,\mu$m radiances by the Visible Infrared Spin Scan Radiometer (VISSR) Atmospheric Sounder (VAS) on board the Geostationary Operational Environmental Satellites (GOES). The $6.7\,\mu$m channel is located near the centre of a strong water vapour absorption band, and under clear sky conditions it is primarily sensitive to the relative humidity averaged over a depth of atmosphere extending from 200 to $500\,$hPa (SB93). The approach of SB93

involved a simplified structure of the atmosphere with linearised profiles of temperature and 1st-order approximations of the Clausius-Clapeyron equation and of the temperature-dependence of the Planck function.

Stephens et al. (1996), in the following SJW96, expanded the research of SB93 to make use of similar data available from polar orbiting satellites. They described a method for retrieving UTH based on the use of radiance data collected by the TIROS Operational Vertical Sounder (TOVS), principally channels 4 ($14.2\,\mu$m), 6 ($13.7\,\mu$m), and 12 ($6.7\,\mu$m) of HIRS. The SJW96

UTH retrieval was based on the SB93 linear formula, with the exception that SJW96 used a varying lapse rate parameter instead of a constant one, as SB93 did, which varied linearly with the difference in HIRS temperature channels 4 and 6 (hereafter $T_4$ and $T_6$, respectively). Few years later, Jackson and Bates (2001), in the following JB01, worked on the retrieval formula of SB93 and determined UTH data using both $T_{12}$ and $T_6$ measurements in the retrieval formula. All these methods regard UTH as a radiance–based quantity, obtained from $T_{12}$ measurements. It is therefore expected that any natural or artificial change in

the $T_{12}$ data will be directly depicted and magnified by the exponential function in the UTH retrievals.

Gierens et al. (2018) speculated that the linearisations involved in the derivation of the traditional retrieval formula would lead to the problems in the low range of $T_{12}$ following the HIRS 2 to HIRS 3 transition. Therefore, we look in this study further into the retrieval formula of UTH and build upon a second–order retrieval formula from HIRS $T_{12}$ data for this important atmospheric quantity. In section 2 we analyse the simple radiative transfer model to derive the usual 1st–order UTH retrievals from

$6.7\,\mu$m radiances and introduce the reader to the second–order approximations. Section 3 presents in depth the derivation of the 2nd–order radiative transfer model UTH retrievals and section 4 tests the 2nd–order model retrievals with some applications. Section 5 provides a discussion of the significance of UTH and sets up a few postulates an UTH retrieval should fulfill in order to be useful and intrinsically consistent. The study ends with the conclusions in section 6.

## 2 Analysis of the first-order retrieval

The relation between the optical thickness between a pressure level $p$ and the top of the atmosphere, $\tau(p)$, and the column density of water vapour above level $p$, $w(p)$, is fundamental to the desired retrieval. SJW96 use an approximation involving the square root of the column density:

$$\tau(p) = k_\lambda \sqrt{w(p)}, \tag{1}$$





where $k_\lambda$ is a spectroscopic constant that depends on wavelength $\lambda$. SJW96 give $k_\lambda = 1.85\,\mathrm{m\,kg}^{-1/2}$ for HIRS 2 and the same value is used here. For HIRS 3 we need a larger value since the optical thickness in channel 12 of HIRS 3 is larger than that of HIRS 2. It turned out that a choice of $k_\lambda = 2.85\,\mathrm{m\,kg}^{-1/2}$ results in a very good comparison between UTHi retrieved from $T_{12}$ measured on common days of NOAA 14 and NOAA 15 operation. That value for $k_\lambda$ is thus chosen for HIRS 3 and HIRS 4.

The column density $w(p)$ for nadir direction is

$$w(p) = \int_0^p \epsilon \frac{r(p')\,e^*[T(p')]}{g\,p'}\,\mathrm{d}p',  \tag{2}$$

where $\epsilon = 0.622$ is the ratio of the molar masses of water vapour and air, $g = 9.81\,\mathrm{m\,s}^{-2}$ is gravitational acceleration, $r(p)$ is the relative humidity profile and $e^*$ is the saturation pressure of water vapour, which depends on temperature and thus indirectly on pressure.

Now, SB93 and SJW96 introduce two linearisations. The first one is for $e^*(T)$:

$$e^*(T) \approx e^*(T_0) \exp\left(\kappa \frac{T - T_0}{T_0}\right).  \tag{3}$$

For $T_0 = 240\,\mathrm{K}$ we have $e^*(T_0) = 37.7\,\mathrm{Pa}$ and $\kappa = 23.1$ (both computed with the vapour pressure formulation of Murphy and Koop, 2005). A similar linearisation is assumed for the relation between pressure and temperature profiles in the upper troposphere with

$$p(T) \approx p(T_0) \exp\left(\frac{T - T_0}{\beta T_0}\right).  \tag{4}$$

Here, $\beta = \mathrm{d}\ln T/\mathrm{d}\ln p \approx 0.22$ is a dimensionless lapse rate.

If we assume a temperature-independent latent heat in the Clausius-Clapeyron equation, the saturation vapour pressure formula can be written as

$$e^*(T) \approx e^*(T_0) \exp\left(\kappa \frac{T - T_0}{T}\right).  \tag{5}$$

This is very similar to the linearised form apart from the denominator in the exponential function. Thus, linearisation is achieved simply by replacing $T$ by the constant $T_0$ which requires that $T$ should be close to $T_0$ where water vapour contributes significantly to the column density.

Similarly, if $\beta = d\ln T/d\ln p$, as it was introduced by SB93, then the correct expression for $p(T)$ must have $T$ instead of $T_0$ in the denominator under the exponential function. However, since $1/\beta$ is much smaller than $\kappa$, deviations of $T$ from $T_0$ have

a smaller effect for $p(T)$ than for $e^*(T)$.

The factor $(T - T_0)/T$ can be developed into a Taylor series:

$$\frac{T - T_0}{T} = \frac{T - T_0}{T_0} - \left(\frac{T - T_0}{T_0}\right)^2 + \left(\frac{T - T_0}{T_0}\right)^3 \dots  \tag{6}$$

Retaining the second order gives already considerable more accuracy to the $e^*(T)$ approximation than the linearised form, that is, we will use now the second-order approximation

$$e^*(T) \approx e^*(T_0) \exp\left\{\kappa\left[\frac{T - T_0}{T_0} - \left(\frac{T - T_0}{T_0}\right)^2\right]\right\}.  \tag{7}$$





Figure 1 shows the ratio of the first and second-order approximations to the correct Clausius-Clapeyron formula for the temperature range 220 to 260 K (we remind that $T_0 = 240$ K). It is seen that the first-order approximation leads to overestimation and errors exceeding ten percent when temperatures below 225 K occur in the layer where channel 12 is sensitive to. The second-order approximation has much smaller errors of less than two percent even at $T = 220$ K. The situation is much better

for the approximation of $p(T)$, and therefore it is not necessary to extend the approximation of p(T) to its second order then.

Also the first order approximation of the temperature dependence of the Planck function,

$$B_1(T) = B(T_0) \exp\left( \frac{hc}{\lambda\, kT_0} \frac{T - T_0}{T_0} \right) \tag{8}$$

is fraught with relative errors of more than 10 percent at low but not untypical upper tropospheric temperatures, see fig. 2. Thus we introduce a second-order approximation in the same way as above:

$$B_2(T) = B(T_0) \exp\left\{ \frac{hc}{\lambda\, kT_0} \left[ \frac{T - T_0}{T_0} - \left( \frac{T - T_0}{T_0} \right)^2 \right] \right\}. \tag{9}$$

For this approximation the relative error does not exceed 3% in the relevant temperature range, which suggests more accurate calculations of brightness temperature data.

## 3   Derivation of the second-order retrieval

### 3.1   Water vapour column density and optical depth

Let us now take the expression for the water vapour column density in eq. 2 and invoke the mean value theorem for integration to draw the profile of relative humidity, $r(p)$, outside the integral and write

$$w(p) = \frac{\epsilon\, \overline{r}(p)}{g} \int\limits_0^p \frac{e^*[T(p')]}{p'}\, \mathrm{d}p'. \tag{10}$$

$\overline{r}(p)$ is a weighted mean relative humidity between the top of the atmosphere and pressure level $p$, defined as

$$\overline{r}(p) = \frac{\int_0^p [r(p')/p']\, e^*[T(p')]\, \mathrm{d}p'}{\int_0^p [1/p']\, e^*[T(p')]\, \mathrm{d}p'}. \tag{11}$$

To proceed with the calculation of column density we need to calculate the integral

$$E^*(p) = \int\limits_0^p \frac{e^*[T(p')]}{p'}\, \mathrm{d}p'. \tag{12}$$

The 2nd-order approximation for the saturation pressure in pressure coordinates is

$$e^*(p) = e^*(p_0) \exp\left[ \kappa\beta \ln\left( \frac{p}{p_0} \right) - \kappa\beta^2 \ln^2\left( \frac{p}{p_0} \right) \right] \tag{13}$$





where we write $p_0$ for $p(T_0)$. Substitution of $x$ for $\ln(p/p_0)$ gives the integral the following simple form:

$$E^*(p) = e^*(p_0) \int_{-\infty}^{\ln(p/p_0)} \exp(\kappa\beta x - \kappa\beta^2 x^2)\,\mathrm{d}x. \tag{14}$$

The integral is of a form

$$\int \exp[-(ax^2 + 2bx)]\,\mathrm{d}x = \frac{1}{2}\sqrt{\frac{\pi}{a}}\exp\left(\frac{b^2}{a}\right)\mathrm{erf}\left(\sqrt{a}x + \frac{b}{\sqrt{a}}\right) + \mathrm{const} \tag{15}$$

(Abramowitz and Stegun, 1972, formula 7.4.32). Inserting the proper expressions for $a := \kappa\beta^2$ and $b := -\kappa\beta/2$, we find

$$\begin{aligned}
E^*(p) &= \frac{e^*(p_0)}{2\beta}\sqrt{\frac{\pi}{\kappa}}\exp\left(\frac{\kappa}{4}\right)\left[\mathrm{erf}\left(\sqrt{\kappa}\beta x - \sqrt{\kappa}/2\right)\right]_{-\infty}^{\ln(p/p_0)} \\
&= \frac{e^*(p_0)}{2\beta}\sqrt{\frac{\pi}{\kappa}}\exp\left(\frac{\kappa}{4}\right)\left[\mathrm{erf}\left(\sqrt{\kappa}\beta\ln(p/p_0) - \sqrt{\kappa}/2\right) + 1\right].
\end{aligned} \tag{16}$$

Using this expression, and putting together all constant prefactors, the vapour column density above $p$ is

$$w(p) = 644.8 \cdot \bar{r}(p)\left[1 + \mathrm{erf}\left(\sqrt{\kappa}\beta\ln(p/p_0) - \sqrt{\kappa}/2\right)\right] \tag{17}$$

and the corresponding optical depth follows from eq. 1. Note that the corresponding prefactor for the retrieval of UTHi is 847.9.

## 3.2   Radiance calculation

The radiance measured by the satellite instrument is given as the solution of the following simple form of the radiative transfer equation (see SB93 and SJW96):

$$I = \int_0^\infty \exp[-\tau(p)]\frac{\mathrm{d}B(p)}{\mathrm{d}p}\,\mathrm{d}p. \tag{18}$$

The second-order formulation of the Planck function as a function of pressure is

$$B(p) = B_0 \exp\left[C_\lambda\beta\ln\left(\frac{p}{p_0}\right) - C_\lambda\beta^2\ln^2\left(\frac{p}{p_0}\right)\right], \tag{19}$$

where we write $B_0$ for $B(T_0)$ and where $C_\lambda = hc/(\lambda k T_0) = 8.95$ is a constant, computed for a wavelength ($\lambda$) of $6.7\,\mu$m (and $T_0 = 240$ K). Further, we have that

$$\frac{\mathrm{d}B(p)}{\mathrm{d}p} = \frac{B(p)\,C_\lambda\beta}{p}\left[1 - 2\beta\ln\left(\frac{p}{p_0}\right)\right]. \tag{20}$$

We now introduce another abbreviation for use in the expression for the optical depth:

$$A_\lambda = k_\lambda\sqrt{644.8}\quad\text{for UTH and}\quad A_\lambda = k_\lambda\sqrt{847.9}\quad\text{for UTHi} \tag{21}$$





and replace $\bar{r}(p)$ with a constant parameter $U$ (or $U_i$) (the value of UTH or UTHi whose meaning will be clarified later). Finally we substitute $x$ for $\ln(p/p_0)$. This gives

$$I = B_0 C_\lambda \beta \int_{-\infty}^{\infty} \exp\left\{-A_\lambda \sqrt{U}\left[1 + \mathrm{erf}\left(\sqrt{\kappa}\beta x - \sqrt{\kappa}/2\right)\right]^{1/2}\right\} \exp[C_\lambda(\beta x - \beta^2 x^2)](1 - 2\beta x)\,\mathrm{d}x. \tag{22}$$

We note here that we also use this equation in the discussion of a formulation of the weighting function, in section 5.3. For
comparison, it will then be useful to abbreviate the integrand as $\Phi(x;U)$, such that $I/B_0 = \int \Phi(x;U)\,\mathrm{d}x$. For comparison we show as well the first-order version that can be derived from SB93:

$$I_1 = B_0 C_\lambda \beta \int_{-\infty}^{\infty} \exp\left(-A'_\lambda \sqrt{U}\, e^{(\beta\kappa+1)\,x/2}\right) \exp(C_\lambda \beta x)\,\mathrm{d}x, \tag{23}$$

where $A'_\lambda$ is an abbreviation for $k_\lambda \left\{(\epsilon\, e^*(T_0)\, p_0)/[g\, p_* (\beta\kappa + 1)]\right\}^{1/2}$. Note that the same substitution has been made here as before for the sake of comparison. Using the logarithm is not actually necessary for the 1st order.

Figure 3 shows the two versions of the integrand just derived. The curves depend as expected on $U$, that is, higher $U$ leads to lower integrals, and thus lower radiances. The variation of the curves is similar in both panels, testifying to the correctness of the second-order calculations. In these integrands, $U$ must be interpreted as that parameter that gives the resulting integral value, the radiance $I$, the correct (i.e. measured) value. The figure shows that for the same value of $U$ the radiances are higher in the 2nd order version than in 1st order.

The necessary integrations have been performed numerically (Romberg integration) and the results for UTH ranging from 1 to 99% are presented in fig. 4 for the 1st and 2nd order retrievals. As expected, the more humid the upper troposphere is, the lower the radiance at the satellite becomes. There is some radiation from altitudes below the $240\,\mathrm{K}$ level when the upper troposphere is quite dry which explains the values above unity. For comparison and test purpose we have included the corresponding result of SB93, directly as given in that paper (labelled "analytically", circles). This curve is practically identical
to the numerically integrated one, which shows that the numerical integration works. The difference between the 1-st and 2nd order formulation turns out to be small but non-negligible.

Finally we need to express the radiances as brightness temperatures, $T_{12}$, where one can exploit that $6.7\,\mu\mathrm{m}$ is on the Wien-wing of the Planck-function at $T_0$, that is:

$$\frac{I}{B_0} = \frac{e^{hc/\lambda k T_0} - 1}{e^{hc/\lambda k T_{12}} - 1} \approx \exp\left[\frac{hc}{\lambda k T_0}\left(1 - \frac{T_0}{T_{12}}\right)\right]. \tag{24}$$

This gives for the channel 12 brightness temperature the expression

$$T_{12} = \frac{T_0}{1 - \ln(I/B_0)/C_\lambda}, \tag{25}$$

where $I/B_0$ is a unique function of $U$. Solving for $U$ leads thus to the desired retrieval function. Four versions of the latter are compared in Fig. 5, viz. the original function derived by SB93 (their eq. 20, constants slightly adapted, that is, $e^*(T_0) = 37.7\,\mathrm{Pa}$ and $\mu = 8.95$, and their eq. 23 which gives a slightly different result), the update of this function by Jackson and Bates (2001)
without their correction for a variable lapse rate (i.e. the prefactor $1/P_H$ they introduced is set to unity), and the new second-order retrieval. The latter gives higher UTH values than the other three retrievals at a given brightness temperature.



### 3.3 The retrieval for different channel central frequencies

The transition from HIRS 2 to HIRS 3 was accompanied by a change of the central wavelength of channel 12 from 6.7 to $6.5\,\mu$m (Shi and Bates, 2011). The spectroscopic quantities changed in a corresponding way. Furthermore, as we are interested in ice supersaturation, it is important to have the retrieval for UTHi as well, which implies different saturation vapour pressure

and different factor $\kappa$. All necessary physical constants are given in table 1 and the resulting retrieval functions are presented in Fig. 6.

Since the atmosphere is more opaque at the latter wavelength, $6.5\,\mu$m, the HIRS 3 sensor is sensitive to a layer that is approximately one kilometre higher than the layer where HIRS 2 is sensitive to. The transition thus implies an average temperature change of $-7\,$K (Gierens et al., 2018). This is reflected in the shift of the retrieval curves (compare the solid with the dotted

UTH curves), the mean difference of which is very close to the expected 7 K. The same stands when comparing the solid and dotted UTHi curves. Therefore we expect that with the new second-order retrievals the number of ice-supersaturated cases from HIRS 2 and HIRS 3 and 4 sensors will become more similar than before. This expectation will be tested in the next section.

The table contains three columns labelled $a, b, c$; these are fit coefficients computed with the Marquart-Levenberg method (Press et al., 1989) that is implemented in `gnuplot` for the retrieval functions. The form of the fit is quite similar to the fit

used by SB93 (eq. 23), apart from the addition of a quadratic term that is appropriate for the 2nd-order retrieval. Thus, for practical purpose one can use:

$$U/\% = 100 \exp\left(a + bT_{12} + cT_{12}^2\right) \tag{26}$$

It is not necessary to show the fits since they are nearly congruent with the retrieval functions.

Finally, we follow JB01 and introduce a prefactor that accounts for variability of the lapse rate factor $\beta$. Instead of repeating

the derivation of the retrieval *ab initio* with variable $\beta$ JB01 introduced a correction that uses the brightness temperature measured with channel 6, $T_6$, and the final result is

$$U/\% = 100 \frac{\exp\left(a + bT_{12} + cT_{12}^2\right)}{a' + b'T_6}. \tag{27}$$

The coefficients $a', b'$ have been recomputed by Gierens et al. (2014) in response to the change of the intercalibration basis of the brightness temperatures between JB01 and Shi and Bates (2011).

## 4 Test applications

### 4.1 Application to computed brightness temperatures

While the jump in the mean retrieved brightness temperatures of channel 12 following the transition from HIRS 2 to HIRS 3 could be remedied with various methods (Shi and Bates, 2011; Gierens and Eleftheratos, 2017; Gierens et al., 2018), a pertinacious problem remained for cases with low brightness temperature (around 230 K and below) and corresponding high

retrieved values of UTHi. We found that much more supersaturation (UTHi$>$100%) and high UTHi in general was retrieved





from NOAA 15 than from NOAA 14 brightness temperatures of the same location and the same day (see for instance fig. 1 of Gierens and Eleftheratos, 2017). Here we show that this problem does no longer occur when the 2nd-order retrieval function for $6.7\,\mu$m is used for NOAA 14 and that for $6.5\,\mu$m is used for NOAA 15. Let us begin first with brightness temperatures derived from radiative transfer calculations for a large set of radiosonde profiles (meteorological observatory Lindenberg, see

Gierens et al., 2018). Here we have two sets of brightness temperatures, one computed for the channel 12 spectral response function of NOAA 14 and one for NOAA 15. We apply now the UTHi fits of table 1 and compare the resulting values. Figure 7 shows that the UTHi pairs from both retrievals are close to the diagonal line and there is not anymore a considerable deviation of the cloud of pairs from the diagonal at the upper end. So it seems that this old problem could be solved with the new retrieval.

### 4.2   Application to the HIRS 2 to HIRS 3 transition

To check the performance of the 2nd–order retrieval formulae on real brightness temperature data, we have used the brightness temperatures measured by HIRS 2 on NOAA 14 and HIRS 3 on NOAA 15 on 1004 common days of operation. These data have been used in recent papers for similar checking purposes, for instance of the cdf-nudging (Gierens and Eleftheratos, 2017) and the superposition methods (Gierens et al., 2018). To perform the check, the brightness temperatures are translated into UTHi according to the new formulation developed in this paper. Then UTHi is gridded on a $2.5° \times 2.5°$ grid and a daily average

is formed for each grid point where data are available. A simple plausibility check removes all data where the corresponding UTH$> 100\%$. The valid daily averages are then compared day-by-day and grid-by-grid. This results in more than 700000 data pairs whose distribution is shown in Fig. 8. As the figure demonstrates, the data pairs are grouped along the diagonal line (black) and there is no deviation from this line towards the high UTHi values. The bivariate regression (red line) has the equation

$$U_i(\text{N15})/\% = 1.17 + 0.998\, U_i(\text{N14})/\% \tag{28}$$

with a slope of practically unity. The mean difference is $-1.3\%$ and the standard deviation is $15.8\%$.

With this result it can be expected that time series of high UTHi cases can be constructed without the need for statistical corrections. This will be considered next.

### 4.3   Time series of occurrence of high UTHi cases

The above mentioned problem of much higher cases of retrieved supersaturation after the HIRS 2 to HIRS 3 transition was first noticed by the authors when we made figure 9a in GE17. That figure was produced using the intercalibrated data of Shi and Bates (2011) and showed a very steep increase in the occurrence frequency of high UTHi ($> 70\%$) cases and the time when the increase started coincided with the start of NOAA 15 and the corresponding transition to a new channel 12 central frequency. Figure 9 shows the time series of high-UTHi-value occurrences from July 1979 to December 2014 computed from

non-intercalibrated HIRS data (monthly frequencies of daily mean values per $2.5° \times 2.5°$ grid box in the midlatitude zone $30°$ to $70°$N) using the 2nd-order retrieval for $6.7\mu$m up to NOAA 14 and for $6.5\mu$m from NOAA 15 on. The figure shows a




tendency for increasing frequencies as well, but it is a mild increase instead of a nearly exponential one as seen in figure 9a of GE17. One can also notice a minimum after 1985 followed by a quite strong increase just before 1990.

For a more detailed study of these time series it would be better to have the series NOAA 6 to NOAA 14 and the subsequent one NOAA 15 to MetOp 2 intercalibrated instead of the here used non-intercalibrated data. For this it seems possible to use

the channel 12 and 6 data intercalibrated to NOAA 12 (Shi and Bates, 2011; Gierens et al., 2014) for NOAA 6 to NOAA 14 and the data that are intercalibrated to Metop-A (Shi and Bates, 2011) for NOAA 15 and later satellites. An intercalibration between NOAA 14 and NOAA 15 should however be replaced by using different retrieval formulae as done here.

### 4.4    The probability distribution of UTHi

The same daily values as before can be used to determine the probability distribution of UTHi. We have done this for the 2nd-

order retrieval with the non-intercalibrated data as above, but for the JB01 retrieval it was necessary to use the intercalibrated data (using JB01 with non-intercalibrated data leads to "interesting" but nevertheless useless results).

Figure 10 shows probability density functions of UTHi in both versions of the retrieval and for different periods of time. Solid curves represent data from 1979–2014, the dashed and dotted curves represent the earliest and latest approximately 10 years in our data set 1979–1989 and 2006–2014, respectively. Recall that in the first of these periods the satellites carried

HIRS 2 only and in the last period only HIRS 3 or HIRS 4. There is no qualitative difference between the curves of the different time periods and HIRS versions. All curves have similar shape but with the 2nd-order retrieval the pdf has a longer tail to high and supersaturated values as can be expected from the foregoing discussion. The upper tail is exponentially distributed in all pdfs shown. An exponential distribution of supersaturation values is expected from many other data sets studied (e.g. aircraft in-situ measurements, radiosondes, other satellites: Gierens et al., 1999; Spichtinger et al., 2002, 2003; Haag et al.,

2003; Gettelman et al., 2006; Lamquin et al., 2009).

Mean values $\pm$ one standard deviation for the pdfs are as follows for the JB01 data: $37.2 \pm 16.7\%$ for the overall data set, $36.2 \pm 15.8\%$ for the period 1979-1989, and $38.5 \pm 17.6\%$ for 2006-2014. The corresponding values for the 2nd-order retrieval are $56.8 \pm 21.9\%$ for all data, $53.6 \pm 21.2\%$ for 1979-1989 and $59.7 \pm 22.6\%$ for 2006-2014. The data show in both retrievals a tendency to more high UTHi and less low values over the long observation period which is consistent to the increasing tendency

that can also be seen in fig. 9. Also the distributions get slightly broader.

## 5    On the meaning and purpose of UTH

### 5.1    Simple postulates

When we are talking about upper-tropospheric humidity we must distinguish the substance water vapour in the upper tropo-sphere from the physical quantity UTH. While the substance is clearly defined and can be seen nicely for instance on satellite

images in the water vapour channels, the definition of the physical quantity UTH is not necessarily unique. If humidity and temperature profiles are given in high resolution, UTH can be defined as a weighted mean over the humidity profile where





the weighting function is given by the transfer of radiation through the atmosphere. Evidently, the definition then depends on radiation details (e.g. wavelength, channel width, filter function, etc.). In practice, however, we don't know the necessary profiles. All that is given is the brightness temperature measured by channel 12, and $U$ is simply a function of $T_{12}$. That there is no UTH per se and thus no true value implies that there is neither a wrong one. Hence the function that maps $T_{12}$ into $U$ is in

principle arbitrary. Is it then possible to validate an UTH algorithm? Now, validation is often mistaken as verification and this makes no sense when there is no truth. Validation, in the sense of Oreskes et al. (1994), means "establishment of legitimacy", which has a connotation of usefulness. Certainly there are more or less useful mappings from $T_{12}$ into $U$ when the interest is a typical or "mean" relative humidity in the upper troposphere. To be a useful quantity, UTH as determined from the brightness temperature should obey a number of postulates.

A simple postulate is: UTH should have values on the familiar scale of relative humidity: that is, $0\% < U < 100\%$ and $0\% < U_i$, where the latter can exceed 100% like RHi does. This is usually fulfilled and values $U > 100\%$ are discarded as bad data.

An increase of relative humidity somewhere in the region to which HIRS channel 12 is sensitive should result in increasing UTH. This postulate is fulfilled if the temperature decreases upwards in the layers where the weighting function (see below)

has non-negligible values. A temperature inversion in that layer would violate the basic assumptions for the retrieval and thus yield bad results (cf. the warning expressed by SB93, after their eq. 16).

Another simple postulate, that is important for consistency, is that in an academic case of a constant relative humidity throughout the atmosphere, say $r(p) = r_0$, $U$ should have the same value, that is, $U = r_0$. This postulate is not generally fulfilled when in the defining integral for $w(p)$ the factor $r(p)/p$ is drawn out, instead of only $r(p)$. The original derivation of

SB93 is equivalent to drawing out $r(p)/p$ of the integral. $p$ is replaced by the constant $p_*$. This leads to

$$\bar{r}(p) = p_* \frac{\int_0^p [r(p')/p'] \, e^*[T(p')] \, \mathrm{d}p'}{\int_0^p e^*[T(p')] \, \mathrm{d}p'}, \tag{29}$$

and the two integrals do not cancel if $r(p) = r_0$, and in general $\bar{r}(p) \neq r_0$. Thus, the consistency postulate is not fulfilled in that case. In contrast, in the approach by SJW96, only $r(p)$ is drawn out, and the two integrals (eq. 9) cancel if $r(p) = r_0$ such that $\bar{r}(p) = r_0$. We followed their method in our derivation.

Furthermore, the ratio $U_i/U$ should depend on brightness temperature of the water vapour channel in a way similar as the dependence of $r_i/r$ on local temperature. This is considered next in some detail.

## 5.2 The relation between UTH and UTHi

Between the local quantities $r(p)$ and $r_i(p)$ (relative humidity with respect to supercooled liquid water and ice, respectively) there holds the relation $r \, e_w^*(T) = r_i \, e_i^*(T)$. A similar relation should hold between $U$ and $U_i$. Fig. 11 shows a num-

ber of curves. The solid black one is the ratio between the saturation vapour pressures vs. temperature (using the formulas of Murphy and Koop, 2005) that holds locally (at a point). The dashed black curve is the corresponding non-local ratio $E_w^*(T)/E_i^*(T)$. Using the fit coefficients from table 1 and eq. 26, we compute $U_i/U$ for the two wavelengths as functions of temperature (red lines). The blue line is the corresponding ratio using the coefficients from JB01. The black rectangle in




the middle represents the range of values we computed from over 1600 radiosonde profiles of Lindenberg (using eq. 9 with the 245 K isotherm as the lower integration limit and eq. 5 for the representation of the vapour pressure). Evidently, the new 2nd-order retrieval leads to $U_i/U$ values that obey the postulation that such ratios should be close to $E_w^*(T)/E_i^*(T)$. $U_i/U$ in the version of JB01 is far below the reference.

This fact explains why in figure 9 the frequency values (y-axis) are much larger than in our previous paper (GE17). In that paper we used the JB01 formulation and this seems to underestimate $U_i$ when it computes $U$ correctly (as fig. 5 shows, the JB01 retrieval of UTH is consistent with the 2nd-order retrieval). Thus it seems that we have underestimated the frequency of high $U_i$ cases in previous papers and the new formulation will give more reliable results.

     We note, however, two subtle issues. First, the retrieved $U_i/U$ ratio depends on the channel wavelength, which is surprising.
The source of the difference lies in the non-locality of this ratio and the fact that $6.7$ and $6.5\,\mu$m photons originate from different layers. The second issue is that $U_i/U$ is not exactly $E_w^*(T)/E_i^*(T)$. This is because the assumed structure of the atmosphere depends a little on whether it is modelled via the relative humidity over liquid water or via the relative humidity over ice. This small difference arises because of the approximation of the saturation vapour pressure formulation. So, in eq. 22 the optical thickness is differently distributed as a function of pressure altitude when we use $r_i$ and the corresponding quantities instead
of $r$. Fortunately, these differences and issues are small.

## 5.3   The weighting function

SJW96 (their eq. 19) introduced the interpretation of UTH as a weighted average over the profile of relative humidity. The weighting function is given by the vertical derivative of the transmission function. In Gierens and Eleftheratos (2016, Appendix A) we derived a generic form for water vapour weighting functions and this form is of double-exponential or Gumbel type (see
also Gierens et al., 2004). We note here a remarkable similarity of this generic weighting function with the integrands of eq. 22 and 23, the latter of which even shares the double exponential form with the generic kernel function. In fact, choosing suitable parameters (scale $h$ and location $x_0$ parameter) it is possible to closely fit $\Phi(x;U)/(I/B_0)$ with a relatively simple Gumbel function of the form

$$\frac{\Phi(x;U)}{I/B_0} \approx h^{-1} e^{(x-x_0)/h} \exp(-e^{(x-x_0)/h}). \tag{30}$$

Fig. 12 gives two examples of this similarity.

     This similarity is still deeper. Both $\Phi$ and the Gumbel-type weighting function have two parts. The double-exponential (or exponential-error-function, $\exp(\cdots \mathrm{erf}(\cdots))$) part in both represents the local value of the transmission function (which in turn depends on the vertical distribution of the absorber, water vapour). The simple exponential in $\Phi$ represents the local emission (Planck function) and the absorption in the weighting function. According to Kirchhoff's law absorption is proportional to
emission. Thus, the integrands and the generic weighting functions share similar shapes and consist of two factors which pairwise share a common or proportionally related meaning. These considerations suggest that UTH is not just a weighted mean over a profile of relative humidity; it is a weighted mean that accounts for the radiative effects of water vapour absorption and emission.





The typical use of the weighting kernel is the computation of $U$ from a profile of relative humidity:

$$U = \int_{-\infty}^{\infty} r(x)\, K(x, x_0, h)\, \mathrm{d}x, \tag{31}$$

where $x = \ln(p/p_0)$ as in eq. 22, and $K$ is the weighting function as given in eq. 1 of Gierens and Eleftheratos (2016). Since $K(x, x_0, h)$ has a form very similar to $\Phi$, we can try the following:

$$U' = \frac{\int_{-\infty}^{\infty} r(x)\, \Phi(x; U)\, \mathrm{d}x}{\int_{-\infty}^{\infty} \Phi(x; U)\, \mathrm{d}x}. \tag{32}$$

The denominator, which is simply $I/B_0$, serves for normalisation since $\Phi$ is not normalised while $K$ is. Note that initially there may be a difference between $U'$ and $U$ on the two sides of this equation. A simple iteration generally suffices to make $U' = U$. As this is achieved, we arrive at an expression for the weighting function where the effect of the humidity distribution on the weighting function itself is explicitly given via the parameter $U$. That is, instead of $K(x, x_0, h)$ one can use $\Phi(x; U)/(I/B_0)$.

Instead of the need for the location parameter $x_0$ which represents the altitude where the optical thickness reaches unity and which must be computed by running a radiative transfer model, the simple iteration and integration (for the normalisation factor $I/B_0$) suffices.

Let's assume that the profile of relative humidity, $r(p)$, is known for a certain case. Then it can be inserted into eq. 32, and a few iteration steps yield both a value of $U$ and of $I/B_0$ or equivalently $T_{12}$. It is as well possible to use $r(p)$ to compute the

profile of $w(p)$, the water vapour column density. If this is given, the transmission function $\exp(-\tau(p))$ can be computed and used directly in eq. 22 and this yields a radiance, hence a brightness temperature as well (from which, in turn, we can compute $U$ using the retrieval formula). Ideally, these two ways of computing $T_{12}$ and $U$ from a given $r(p)$ yield the same results. We have checked this using realistic profiles of relative humidity from the Lindenberg radiosonde data set as above. The result is presented as the difference of the two versions of $T_{12}$ in Fig. 13. The figure shows that the comparison typically yields

differences of about 1 K, which imply differences in UTH of 4-6% depending on $T_{12}$ (cf. fig. 5). Sometimes the differences are much larger, for instance for the profile from 9 May 2000, 18 h (no. 329). This is a bad profile that ceases at about 400 hPa. Other deviations come from the simple trapezoidal integration for $w(p)$ that we used here instead of the Romberg method, and in particular from the fact that a transmission profile computed via an actual $w(p)$ generally differs from the idealised one that must be used for the retrieval (the first factor in $\Phi(x; U)$. Nevertheless, the differences are quite small.

Here is therefore the recipe for getting both the desired value $U$ and an estimate of the weighting function. First, one determines $U$ via the retrieval formula and, then, the weighting function is approximately $\Phi(x; U)/(I/B_0)$.

## 5.4 The ill-posedness of the problem and interpretation of UTH

Nothing demonstrates the ill-posedness of the problem to determine a typical relative humidity of the upper troposphere from just one datum, the measured brightness temperature, better than eq. 22. On the left side of the equation there is the given

radiance (measured or computed with a radiative transfer model) which one tries to reproduce with an appropriate choice of $U$ in the function $\Phi$. Now, while $I$ (or $T_{12}$) is the result of a full radiative transfer through the real atmosphere, $\Phi$ represents





a simplified radiative transfer through a pseudo-atmosphere. One must conclude that $U$ is the mean UTH in such a simple atmosphere with simple radiation physics that would yield the same radiance as the true measured one.

Furthermore, eq. 22 is not the only possibility for a simplified radiative transfer in a pseudo-atmosphere. Eq. 23 shows an example of the 1st-order approximations. Evidently, using eq. 23 instead of 22 requires a different $U$ to reproduce the measured

radiance. But this is still not the end of the story; the operator $\Phi(x; U)$ can represent any model atmosphere and any more or less sophisticated radiative transfer. This combination determines the resulting $U$. In this spririt it is possible to obtain an optimal $U$ if actual profiles of $r(p)$ and $T(p)$ are given and if $\Phi(x; U)$ represents a full radiative transfer through the actual atmosphere. In this case, $\Phi$ should be written as $\Phi[p; r(p), T(p)]$ to make the dependencies clear.

It is evident that UTH (and UTHi) does not only represent the humidity in the upper troposphere and the radiation transfer as

such. It also represents the kind of model atmosphere and the kind of radiative transfer model that is used to derive the operator $\Phi$.

## 6   Conclusions

In spite of the largely successful intercalibration of the HIRS channel 12 data (Shi and Bates, 2011), which works well for the bulk of the data, there remained a pertinacious problem in the high range of upper-tropospheric humidities retrieved from

the intercalibrated brightness temperatures: the change in channel wavelength from $6.7\,\mu$m to $6.5\,\mu$m where the atmosphere is more opaque resulted in a quite strong increase in the frequency of high and very high values of UTHi. Over the last years we tried different potential remedies (adjustment of low brightness temperatures using so-called cdf nudging, and additional use of channel 11 brightness temperatures), but these were not completely satisfying. Gierens et al. (2018) speculated that the linearisations used in the derivation of the original retrievals by SB93 and SJW96 were insufficient for the tails of the UTH and

UTHi distributions and we hoped that a retrieval with second-order approximations would yield better results. In reworking the derivation we noticed the necessity to adapt the optical constant $k_\lambda$ to the changed channel wavelength and it turned out that this adaptation is the key to the desired smooth transition from HIRS 2 to HIRS 3, that is a transition without a jump in the frequency of high UTHi values.

The achievements of this paper are:

1. The second-order approximations give a more accurate representation of the variation of water vapour saturation pressure and the Planck function with temperature in the relevant range of $T$ than linearisations.

     2. The change in $k_\lambda$ reflects the change of wavelength and leads to a smooth HIRS 2 to HIRS 3 transition. Comparing data from 1004 days of common operation of NOAA 14 and NOAA 15 leads to a scatter plot of pairs UTHi values that has a bivariate regression function with a slope of nearly unity and small intercept.

3. For future applications we provide retrieval coefficients for both UTH and UTHi, and for both HIRS 2 and HIRS 3/4. The mathematical form of the retrieval formula is similar to that given by JB01, but including the second order of $T_{12}$ under the exponential function.





4. A first application of the new method indicates a moistening of the upper-troposphere (in northern midlatitudes) over the last decades, consistently with an earlier result (Gierens et al., 2014).

5. We argue that our method fulfills a set of consistency requirements (postulates).

6. We derive a new method to estimate the weighting function.

5 The second-order retrieval has been used to produce UTHi data for the northern mid-latitudes. The same retrieval will be applied to produce UTHi data for the tropics (30°N to 30°S) and the southern mid-latitudes (30°S to 70°S). A near-global UTHi satellite dataset will allow us to investigate the natural variability of UTHi on larger space scales. We need to know whether natural cycles or other events (e.g. seasonal variations, Quasi Biennial Oscillation, El Nino Southern Oscillation, North Atlantic Oscillation, solar cycle) have an imprint on UTHi that can be detected in the data after application of our

10 retrieval, and to understand to what degree these natural fluctuations affect the natural variability of UTHi. With this long-term dataset of 35+ years, we are particularly interested to explore trends in the mean and in high UTHi values, which are important to follow the evolution of cirrus cloudiness in a future warmer climate. The studies are going to be performed using a time series that is composed of two intercalibrated datasets, one for HIRS 2 only, and one for HIRS 3 and 4, as well as one without an intercalibration between NOAA 14 and 15 as has been done here.

## 7 Code availability

Does not apply.

## 8 Data availability

HIRS data can be retrieved from NOAA web services.

*Author contributions.* KG made the calculations, KE checked them. Both authors wrote the text.

20 *Competing interests.* The authors declare no competing interests.

*Acknowledgements.* The authors are grateful for the pioneering work by B. Soden, F. Bretherton, G. Stephens, D. Jackson, I. Wittmeyer, and J. Bates. Without their ideas the present work would not have been possible. We are grateful to NOAA for the many years of collection and furnishing of the data that we urgently need to learn more on climate change and how it impacts the humidity structure of the atmosphere. We thank Margarita Vazquez-Navarro for critical reading of an early version of the manuscript.





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



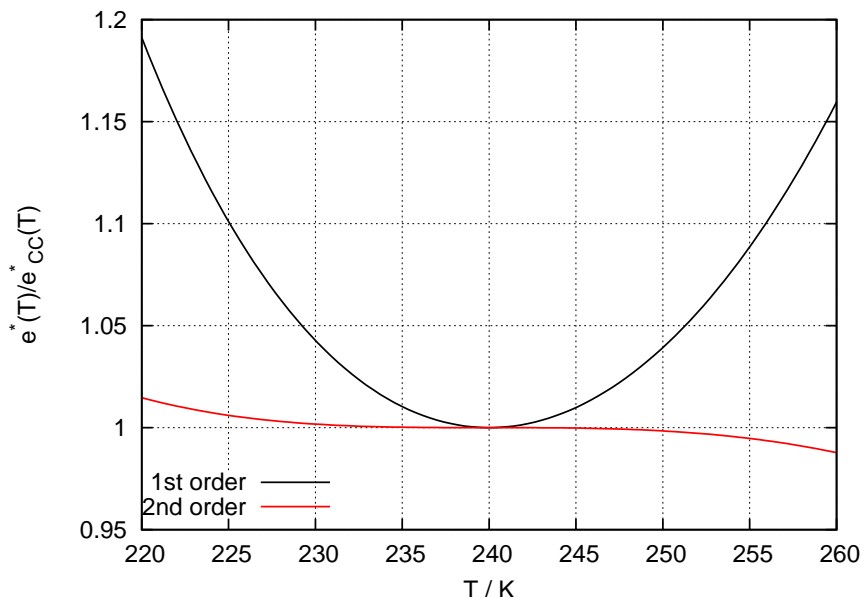

**Figure 1.** Ratio of the 1st-order (black) and 2nd-order (red) approximations to the correct Clausius-Clapeyron saturation pressure of water vapour.

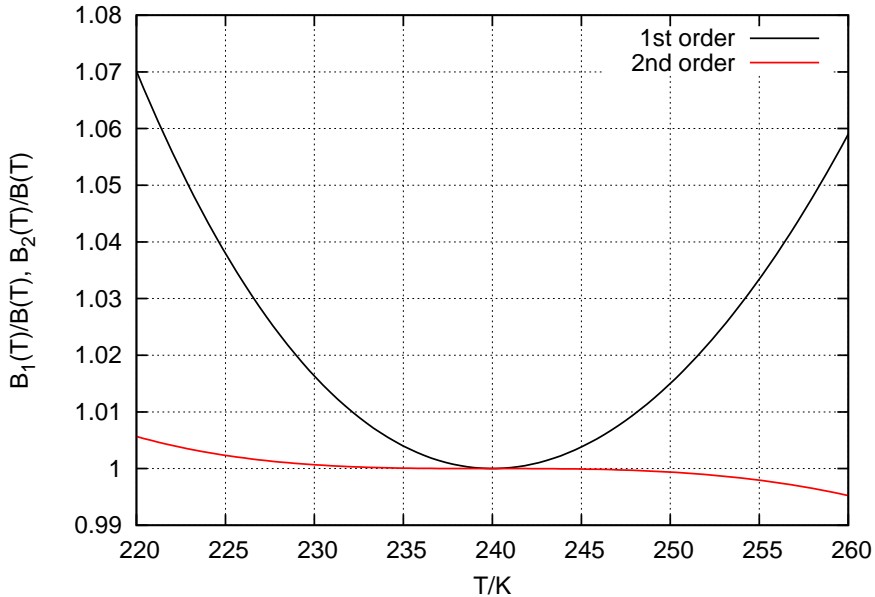

**Figure 2.** Ratio of first (black) and second order (red) approximations of the Planck function to the true Planck function vs. temperature, $T$. The wavelength for the calculation of the Planck functions was $6.7\,\mu$m.





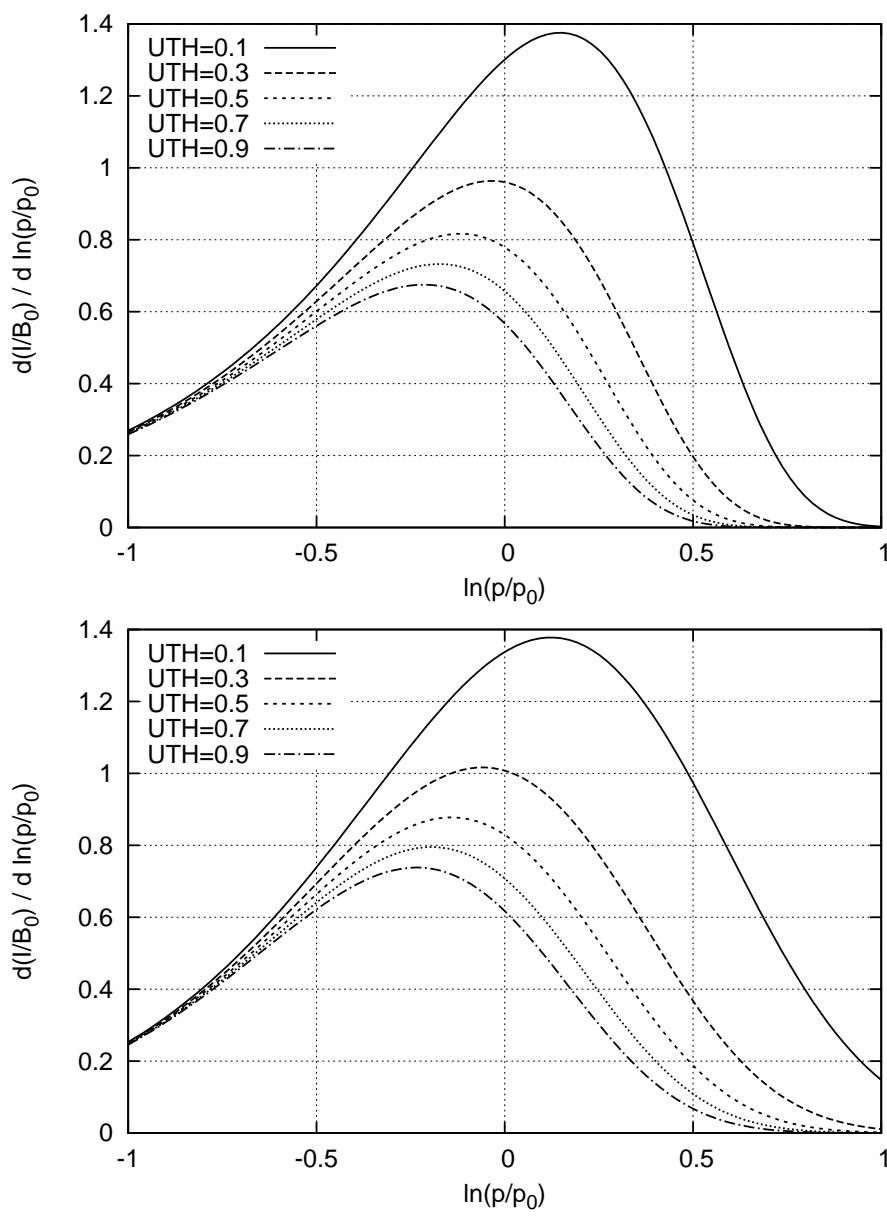

**Figure 3.** The function (normalised by the Planck function at $T_0$) to be integrated to get the radiance in channel 12 as a function of $\ln(p/p_0)$ and shown for various values of the UTH. Top: using the first-order approximations of SB93. Bottom: using the second-order approximations (this study). The higher UTH the less radiation is received at the satellite.



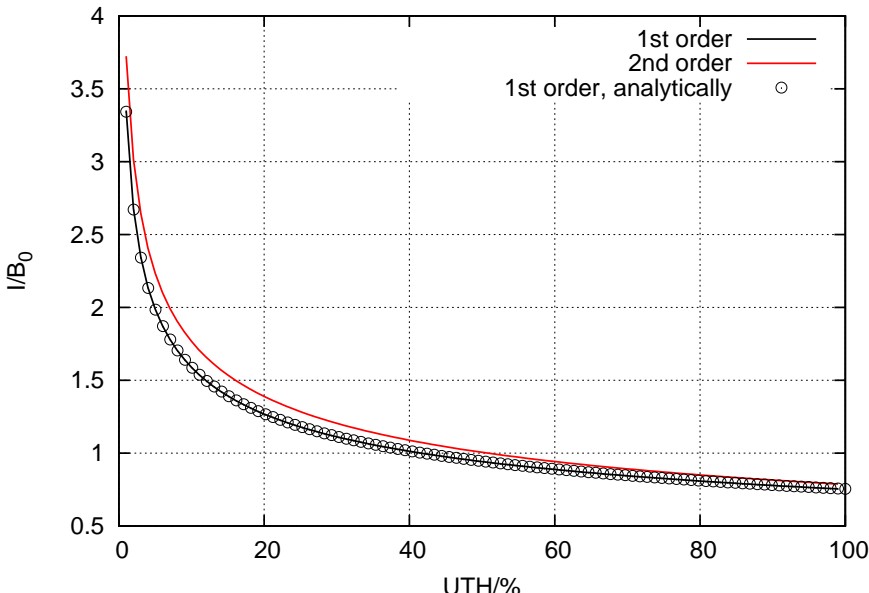

**Figure 4.** Normalised radiances, $I/B_0$, resulting from numerical integration of equations 22 and 23 for the first (black) and second-order (red) retrievals. The circled curve is computed directly from the analytical formula (eq. 18 of SB93).

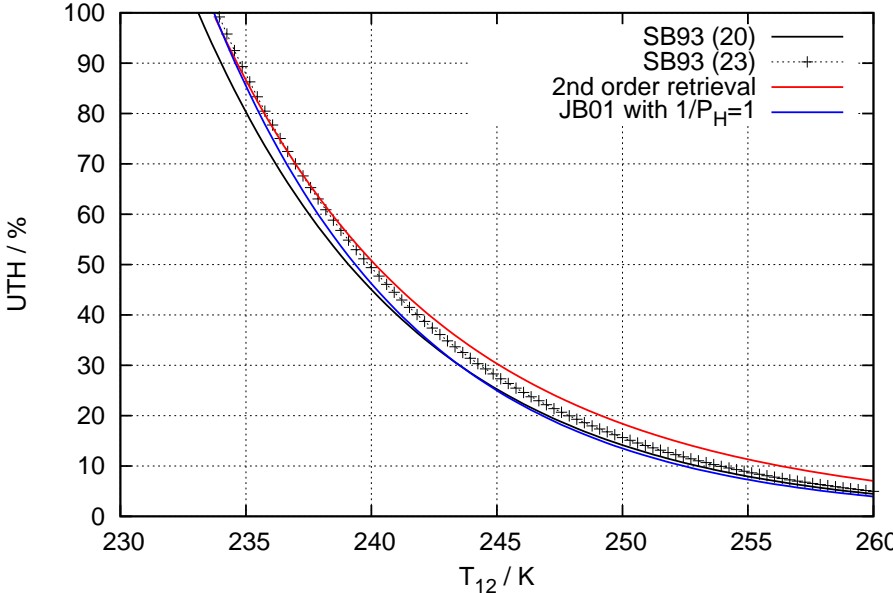

**Figure 5.** Comparison of retrieval functions UTH vs $T_{12}$: The simple black line is the original function from SB93 (eq. 20), the black line with crosses is their eq. 23. The blue line is the improved SB93-type retrieval from JB01, shown here without the correction for lapse rate variability (i.e. $1/P_H = 1$), and the red curve represents the second-order retrieval.



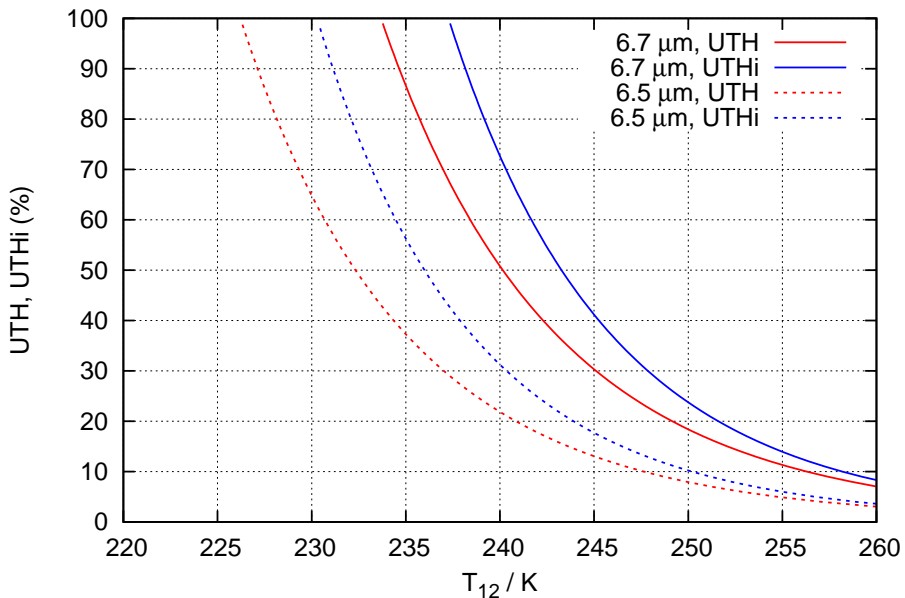

**Figure 6.** 2nd-order retrieval functions for all cases of table 1.

**Table 1.** Constants used for the two different central wavelengths of HIRS channel 12 and fit coefficients.

|  | $e^*$ [Pa] | $\kappa$ | $\lambda$ [$\mu$m] | $A_\lambda$ | $hc/\lambda k T_0$ | $a$ | $b$ [1/K] | $c$ [1/K$^2$] |
|---|---|---|---|---|---|---|---|---|
| UTH |  |  |  |  |  |  |  |  |
|  | 37.7 | 23.1 | 6.7 | 46.98 | 8.95 | 43.36 | $-0.2619$ | $3.266 \cdot 10^{-4}$ |
|  | 37.7 | 23.1 | 6.5 | 72.37 | 9.22 | 45.50 | $-0.2868$ | $3.784 \cdot 10^{-4}$ |
| UTHi |  |  |  |  |  |  |  |  |
|  | 27.3 | 25.7 | 6.7 | 53.87 | 8.95 | 47.69 | $-0.2846$ | $3.522 \cdot 10^{-4}$ |
|  | 27.3 | 25.7 | 6.5 | 82.99 | 9.22 | 50.05 | $-0.3109$ | $4.063 \cdot 10^{-4}$ |





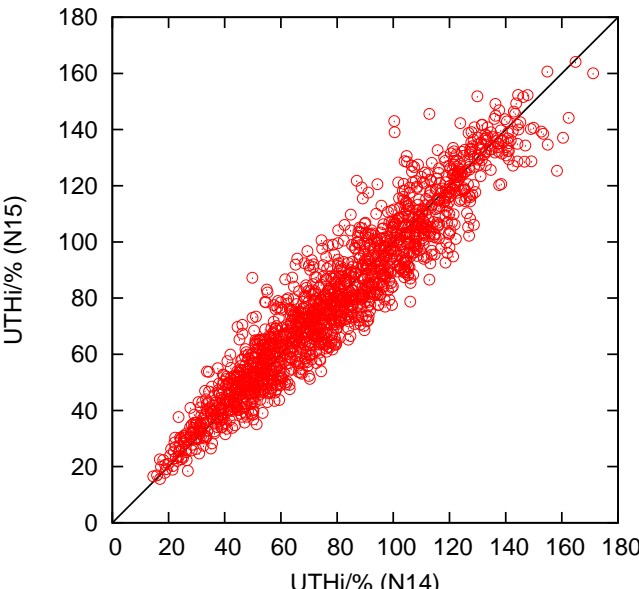

**Figure 7.** UTHi values from channel 12 brightness temperatures computed with a radiative transfer code using the NOAA 14 and NOAA 15 spectral response functions (Gierens et al., 2018). The UTHi values have been computed with the fit function given in the text using the coefficients from table 1 for the two central wavelengths 6.7 (N14) and $6.5\,\mu$m (N15). The black line is simply the diagonal and all points would ideally lie above it. Contrary to earlier intercalibration attempts, the problem of having much more supersaturation cases in NOAA 15 than in NOAA 14 data is no longer present.





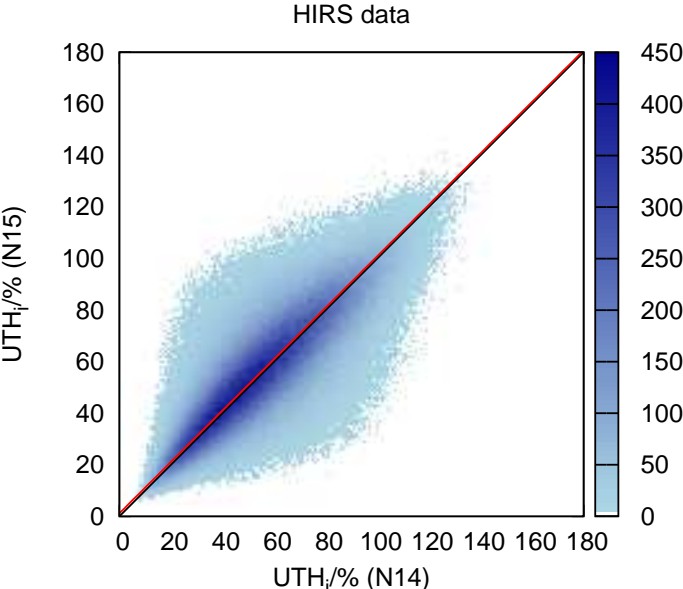

**Figure 8.** Heatmap of UTHi from NOAA 15 vs UTHi from NOAA 14 for 1004 common days of operation. The corresponding brightness temperatures have been transformed into UTHi using the 2nd-order retrieval formulae derived in this paper. More details are given in the main text. Evidently the data pairs are nicely spread out along the diagonal line and there is not anymore the deviation from the diagonal in the upper UTHi range as before. The red line represents the bivariate regression between the two data sets. It is almost identical to the black diagonal.



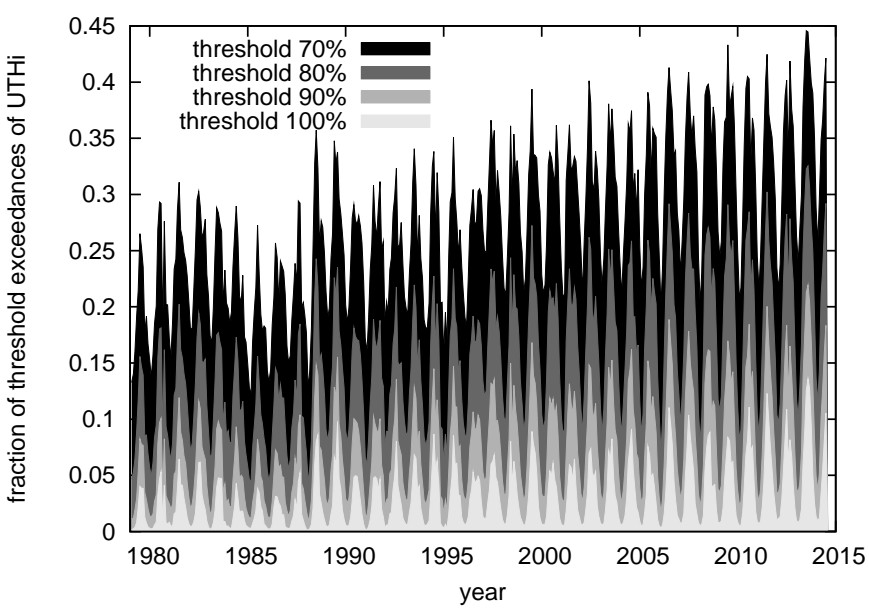

**Figure 9.** Timeseries of monthly fractional threshold exceedances for UTHi exceeding 70, 80, 90, and 100% (various grey scales) in the northern hemisphere between 30 and 70°N and from July 1979 to December 2014.



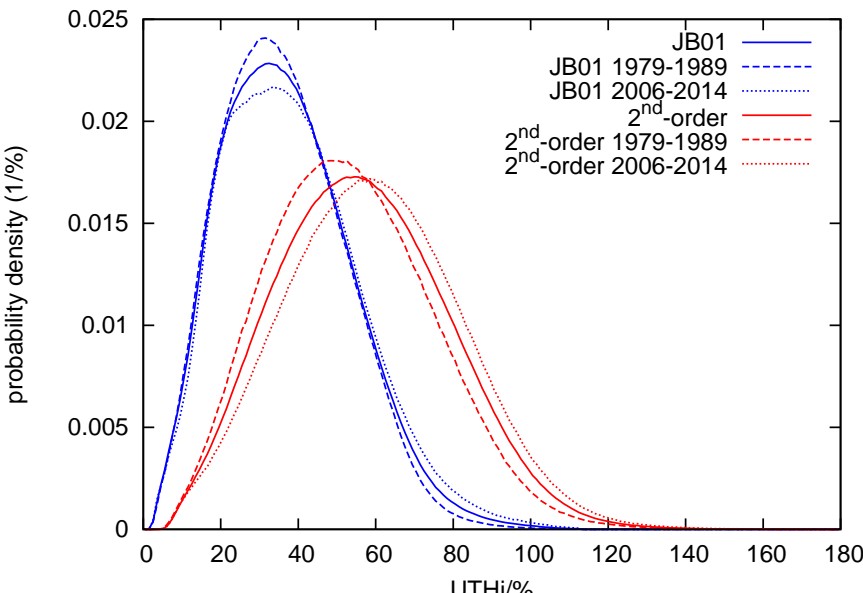

**Figure 10.** Probability density functions of UTHi retrieved with the formula of JB01 (using intercalibrated brightness temperatures, blue) and with the 2nd-order retrieval (using non-intercalibrated brightness temperatures, red). Solid curves represent data from 1979-2014, dashed: data from 1979-1989, dotted: data from 2006-2014. We note that all curves have similar shape but with the 2nd-order retrieval the pdf has a longer tail to high and supersaturated values. The upper tail is exponentially distributed in all pdfs shown. The data show in both retrievals a tendency to more high UTHi values over the long observation period.





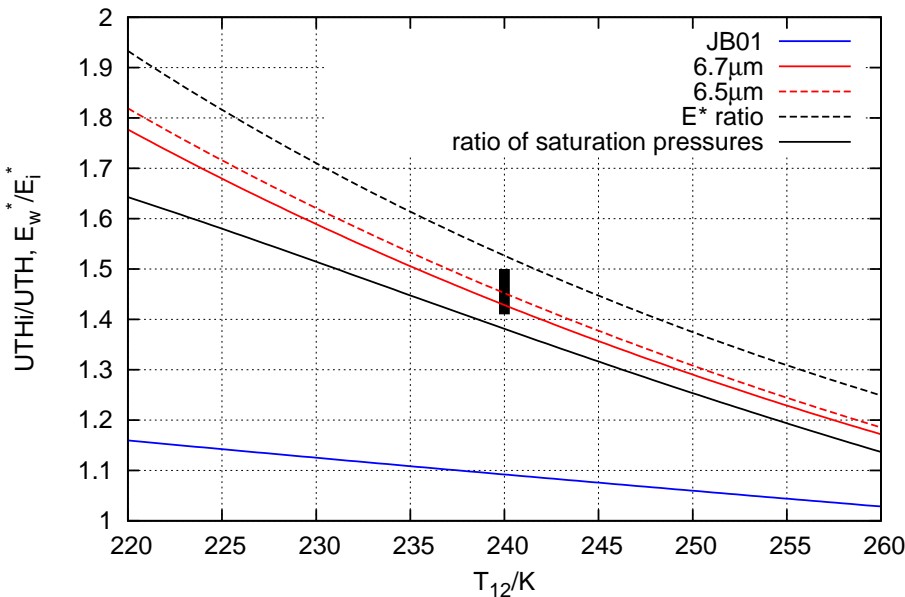

**Figure 11.** Ratio of UTHi vs UTH retrieval (blue: JB01, red: present paper using the retrieval for $6.7\,\mu$m (solid) and the same but for the $6.5\,\mu$m retrieval (dashed)) as function of brightness temperature. The other curves are for comparison. The black solid line is the ratio of saturation vapour pressures $e_w^*(T)/e_i^*(T)$ (Murphy and Koop, 2005) and the black dashed line represents the ratio of the corresponding $E_{w/i}^*(T)$ functions (Eq. 16 with the corresponding parameters from table 1). The black bar in the middle represents more then 1600 values computed by integration over the actual RH, RHi and $T$-profiles from the Lindenberg radiosoundings, assuming the lower integration limit is 245 K (eq. 9). Obviously, the 2nd-order formula meets the actual data quite well, but the ratio obtained from the JB01 coefficients is way off.





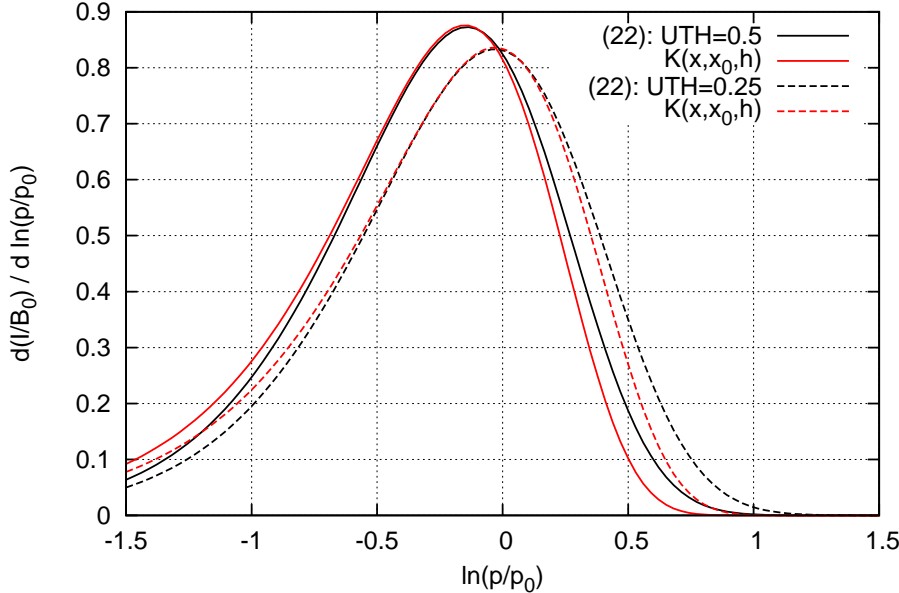

**Figure 12.** Comparison of the normalised integrand of eq. 22 (black) with a generic weighting function of double-exponential (or Gumbel-) type (red).

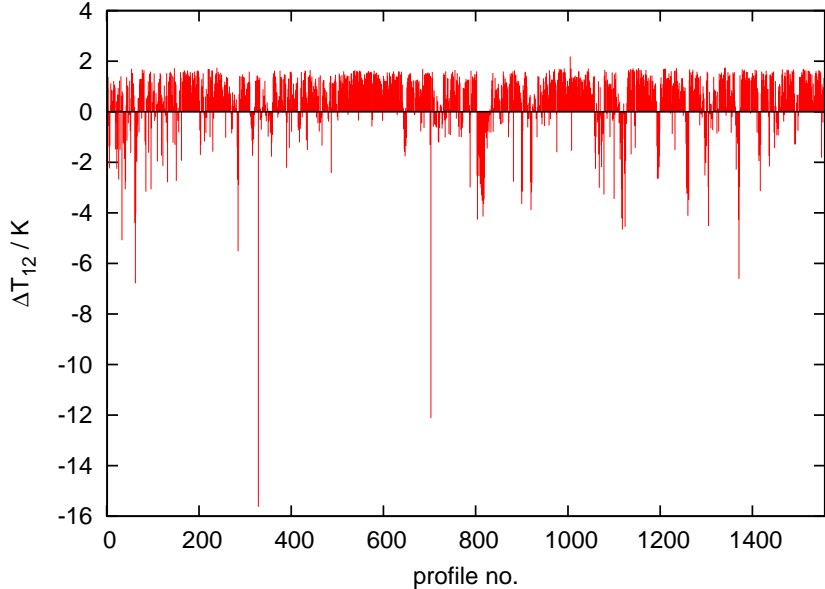

**Figure 13.** Comparison of brightness temperatures computed using actual profiles of relative humidity and water vapour column densities. The computation is either via the simple radiative transfer that is used for the retrieval or via the iterative procedure using the weighting function that is described in the text. Large deviations (more than about 3 K) arise from bad profiles.