# Peer review of "On the interpretation of upper-tropospheric humidity based on a second–order retrieval from infrared radiances"

_Atmospheric Chemistry and Physics, 2018_

## Referee Comment (RC1) · Anonymous Referee #2 · 11 Dec 2018

This paper discusses the definition of Upper Tropospheric Humidity (UTH), a quantity derived from 6.3$\mu$m observations, and which has drawn a lot of attention since the 80s. This specific study looks at the retrieval formula itself, first analytically determined by Soden & Bretherton in 1993. I understood that the scope is to refine the retrieval to reach a better homogeneity in the HIRS2-HIRS3-HIRS4 database, and more precisely for the cold temperature, where the UTH is over saturation (with respect to ice).

However, while reading the paper, I didn't clearly catch its added-value. After reading it several times, I am sorry to say that I still don't see it. In my opinion, several aspects have been eluded: from the detection of clouds within the brightness temperature measurements in the water vapour channel, to the retrieval itself and its application. I have then several remarks that will probably show that I despite I believe that such work on

the interpretation of "water vapor" channel observation is very interesting, the present work needs to be more clear on its roots, its dataset, and its objectives.

**1. The retrieval of UTH from 6.3$\mu$m observations is only possible when there is no clouds in the free troposphere. Brogniez and co-authors discussed at length that point in a 2006 study, where they show that clouds below ~700hPa are almost "unseen" in that channel. Here the study focuses on the cold range of the 6.3$\mu$m temperatures. ïČř Can the authors discuss the cloud screening technique? Can't the very low brightness temperatures mentioned (230K and below) be reasonably associated to partially cloud-covered pixels? In their 2016 work, Chung et al mention the problem of the cloud detection in the HIRS serie which is done thanks to ISCCP. ïČř Therefore, my question here is: would that be possible that the discrepancy in the cold range values would be induced by the cloud screening (and thus ISCCP cloud detection), from HIRS2 to HIRS4?**

**2. When reading UTH-related papers I am always stunned that people don't give credit where credit is due. I agree that B. Soden & co-authors developed the first analytic retrieval of UTH, but the retrieval of UTH from 6.3$\mu$m observations was first developed by J. Schmetz and O. Turpeinen in 1988 and produced operationally at EU-METSAT ever since. I deeply regret such evolution in the referencing. It is obvious that every paper cannot list all the previous work performed on a particular topic, but then when the paper goes back to the roots, then I totally believe that this has to be done. Hence, a whole part of UTH retrieval has been put aside, and more precisely on the work on METEOSAT data: following Johannes Schmetz work, you have also Brogniez et al 2009 and (maybe closer to us) Schröder et al 2014 that have redefined the weighting function by showing that the transmission-derived weighting function of SB93 was the least accurate one, that didn't consider the radiatively-driven information, as the authors underline it. But then the part 5.3 (as well as 5.4. . .) is, in my opinion, a little bit "reheated". . . => Can the author go through the definition proposed in Schröder's work for instance and compare to their definition of the weighting function?**

**3 Section 2, present the 1st order retrieval, as designed by SB93 and SJW96. => Eq (1) and (2) are from the Malkmus band model, adapted to strong absorbing lines, not specified. Since this section intends to re-discuss the SB93 & SJW96 works, then all the assumptions need to be written. => It is nowhere specified that the developments by SB93 and SJW96 are adapted for tropical and subtropical standard atmospheres: in these regions the temperature profile doesn't change much and that is why the 6.3$\mu$m observations can be interpreted as UTH. However, the authors have applied it to northern mid-latitudes. I would like to know how the approximations, and more specifically the linearizations, translate from the tropics to the mid-latitudes. For example, the pressure at which T0 =240K varies slightly between the tropics and the mid-latitudes. Is there an impact?**

---

## Referee Comment (RC2) · Anonymous Referee #3 · 31 Dec 2018

The paper describes a new method to retrieve upper tropospheric humidity (UTH) based on HIRS channel 12 brightness temperatures in an effort to bridge the channel's central wavelength change from 6.7 $\mu$m to 6.5 $\mu$m between HIRS2 and HIRS3 instruments. Built upon the work of past studies, this study retains the second order term in the linearization of the UTH formulation. The addition of the second order term and the choice of the spectroscopic constant for the retrievals are shown successfully matching the UTH between HIRS2 and HIRS3 observations. I am glad to see the resulting good match between HIRS2 and HIRS3 UTH. I think that the study falls into the scope of ACP. The paper is generally well written.

Specific Comments and edits:

Page 1, line 4: Define UTHi.

[Figure]

Page 2, lines 18-19: HIRS3 instruments are flown on N15, N16, and N17. HIRS4 instruments are flown on satellites after.

Page 4, line 3: The choice of optical constant needs a lot more explanation here. How was the optical constant = 2.85 derived? How sensitive is the constant to the retrieval?

Page 8, line 23: It'd be helpful to include the values of a' and b' here so readers don't have to find them from the earlier paper.

Page 20, Figure 5: It is not clear which equation was used for the 2nd order retrieval line. It'd be helpful to include the equation in the caption, or make it clear in the text. On page 7 the text only mentions "Solving for U leads thus to the desired retrieval function".

Page 25, last line: Change "more high" to "higher".

---

## Referee Comment (RC3) · Anonymous Referee #1 · 8 Jan 2019

**General Comments:**

This paper updates an upper tropospheric humidity (UTH) retrieval method introduced by Soden and Bretherton (1993) using the GOES infrared imager channel and was later extended to the HIRS 6.7 micron water vapor channel on NOAA polar orbiting satellites. The purpose for the update was to better mitigate a known bias known to exist in the HIRS UTH retrieval due to a shift in central frequencies between the HIRS/2 and HIRS/3 instruments. The introduction of a second order term in the retrieval formulation is shown to significantly reduce the bias between these instruments especially for retrievals conducted using relatively cold brightness temperatures. The updated retrieval is applied to the NOAA HIRS observations over a 35 time period and it is found that UTHi has a positive trend over that period.

The mathematical description describing the updated retrieval is described well and the addition of the 2nd order term to the linearized equation is shown to provide an improvement to the retrieval formulation. However, there are sections where the text should be made more concise and my suggestions are given in the minor corrections section. I feel the topic of this paper meets the requirements of the ACP journal in that the retrieval is has global extent, it describes a physical process of the troposphere, and it uses a remote sensing technique to identify a gas (water vapor) in the troposphere.

One issue I have is that the title of the paper needs to be more specific. I think the title should indicate that this is an update to an existing HIRS UTH retrieval which only involves retrievals of infrared observations.

I also have concerns that there is no description on the cloud clearing technique used to derive the UTH data in this paper. Cloud contaminated observations will generally reduce brightness temperature and which will likely cause a positive UTH bias in the retrieval should they be introduced. The positive trend in UTH shown in this paper should be clearly identified as a trend derived from clear-sky observations. Furthermore, cloud contaminated satellite observations could contribute to that trend depending on how stable the cloud-cleared technique is for these observations are over the data record.

I also think the positive trend in UTH shown in figure 9 and discussed in item 4 in the conclusions needs to include another caveat to the discussion. That issue is the possible effects of diurnal bias from HIRS observations on the trend in the time series. Diurnal bias is an issue with most NOAA polar orbiting satellites since observation times drift through the diurnal cycle.

Minor Corrections and Comments:

I suggest changing all references to HIRS2, HIRS3, and HIRS4 to the more commonly used acronyms of HIRS/2, HIRS/3, and HIRS/4.

Satellite names interchange between the NOAA-14 and N14 format throughout the paper. I suggest only using one name type.

Page 2, Line 1: I suggest changing "... and study with a view to determining ..." to "... and study for determining ..."

Page 2,Line 11: Polar-orbiting satellite sensors only have this advantage.

Page 2, Lines 11-13: I suggest a more concise sentence – "For climate variability studies it is important to understand the continuity of long term measurements in both the stratosphere and upper troposphere".

Page 2, Line 17: The first HIRS/2 instrument was on TIROS-N which was the satellite before NOAA-6.

Page 2 Line 22: Replace "an unwanted" with "a".

Page 2, Line 24: Replace "solved" with "corrected"

Page 3, Line 22: I suggest changing "we look in this study further into the retrieval formula" to "we further studied the retrieval formula"

Page 5, line 8: I suggest changing "is fraught with relative errors" with "has relative errors"

Page 8, line 13: I suggest changing "The table" to "Table 1"

Page 9, Line 2: I suggest changing "this problem does no longer occur when" to "this problem no longer occurs when"

Page 9, line 8: I suggest changing "there is not anymore a considerable derivation" to "there is little deviation"

Page 10, line 1: I suggest changing "mild" to "small"

Page 10, line 4: I suggest changing "instead of the here used non-intercalibrated" to "instead of using non-intercalibrated"

СЗ

Page 10, line 11: The statement in parenthesis should be omitted unless more specific detail is provided on what interesting but useless results means.

Page 10, line 24: suggest changing "more high UTHi and less low values" with "higher UTHi and fewer low values"

---

## Author Comment (AC1) · 6 Feb 2019

**Reply to the reviewer's comments on the Manuscript No. ACP-2018-1129
by K. Gierens and K. Eleftheratos**

**Remarks**

We thank the reviewers for thorough reading of the manuscript. In the following, reviewer comments are copied in italic font, our replies are in upright font. Essential changes in the manuscript are printed in red colour.

**1 Reply to reviewer 1**

**1.1 Title of the paper**

*One issue I have is that the title of the paper needs to be more specific. I think the title should indicate that this is an update to an existing HIRS UTH retrieval which only involves retrievals of infrared observations.*

We agree and change the title into "On the interpretation of upper-tropospheric humidity based on a second–order retrieval from infrared radiances".

**1.2 Cloud clearing**

*I also have concerns that there is no description on the cloud clearing technique used to derive the UTH data in this paper. Cloud contaminated observations will generally reduce brightness temperature and which will likely cause a positive UTH bias in the retrieval should they be introduced. The positive trend in UTH shown in this paper should be clearly identified as a trend derived from clear-sky observations. Furthermore, cloud contaminated satellite observations could contribute to that trend depending on how stable the cloud-cleared technique is for these observations are over the data record.*

This question refers only to the detected moistening trend. In fact, cloud influence can in principle feign such a trend if thin, non-detected cirrus clouds (or contrails) have an increasing trend themselves. This is possible, but improbable. We have checked the Aura Microwave Limb Sounder UTRHI data (upper tropospheric relative humidity with respect to ice) and found an increasing trend as well for the northern mid-latitudes (but for a short time period only, 2005-2018). Microwave data are insensitive to thin clouds, hence a trend in UTRHI can hardly be feigned by increasing thin cirrus. Furthermore there are publications (Eleftheratos et al., 2007; Minnis et al., 2004) that show decreasing cirrus trends at many locations of the world except for regions with strong air traffic where contrails balance the decreasing cirrus. Of course, contrail cirrus (old widely spread out contrails) is probably increasing since air traffic increases. But Figure 3 (bottom) of Gierens et al. (2014) shows the largest UTHi increase over Siberia and a decrease over the western part of the north Atlantic. That is, an increase where air traffic is still low and a decrease where air traffic is dense and has strong growth rates. Thus, cloud influence on the increasing UTHi trend is possible, but not supported by other sources of evidence.

Most of these points are now added to section 4.4.

**1.3 Diurnal bias**

*I also think the positive trend in UTH shown in figure 9 and discussed in item 4 in the conclusions needs to include another caveat to the discussion. That issue is the possible effects of diurnal*

*bias from HIRS observations on the trend in the time series. Diurnal bias is an issue with most NOAA polar orbiting satellites since observation times drift through the diurnal cycle.*

Also this question refers to the detected UTHi trends. We find it difficult to imagine how the drift of the satellite overpass times can feign such a trend. A trend can certainly be produced when there is only one satellite combined with a diurnal cycle of upper-tropospheric relative humidity that is stable over the whole observation period. But this is not what we have here. Typically there are two or three NOAA satellites simultaneously in orbit with different overpass times (e.g. one in the morning and one in the afternoon). The drifts of simultaneous satellites already cause a mixture of feigned trends (if there is a stable diurnal humidity cycle), perhaps with a small residual trend. Then, all satellites in the NOAA series drifted and they drifted independently from each other (different directions and with different rates). This implies than that the small residual pseudo-trends mentioned above are sometimes positive and sometimes negative. The overall pseudo-trend should be very small then because single trends get averaged out. Finally, trends differ in different regions (see again Figure 3 in Gierens et al., 2014) and this cannot at all be explained by orbital drifts. Summarising, we deem it very implausible that orbital drifts could feign a decadal trend in any quantity.

There is still another possibility: it might be possible that the influence of the global CO2 increase on $T_6$ could perhaps contribute to the trend. Shi et al. (2016) computed that a CO2 increase from 330 to 410 ppmv would on average lead to a decrease in $T_6$ by $2\,K$. Gierens et al. (2014, their Figure 7) also looked at the decadal change in $T_6$ and found it to not undercut $-1\,K$ and to be statistically insignificant. But still it might be physically significant and contribute to the observed UTHi increase. These considerations have now been added to section 4.4 as well.

**1.4   Minor points**

- We use now the form HIRS/2/3/4 throughout the text.

- We changed to the form NOAA nn (except for labels in figures and the regression equation).

- We followed all additional suggestions by the reviewer.

**2   Reply to reviewer 2**

*… while reading the paper, I didn't clearly catch its added-value. After reading it several times, I am sorry to say that I still don't see it. In my opinion, several aspects have been eluded: from the detection of clouds within the brightness temperature measurements in the water vapour channel, to the retrieval itself and its application. I have then several remarks that will probably show that I despite I believe that such work on the interpretation of "water vapor" channel observation is very interesting, the present work needs to be more clear on its roots, its dataset, and its objectives.*

The added value of this study is the formulation and application of the second-order retrieval for UTH, which, to our knowledge, has not been examined until now. Building upon the work of past studies, we have formulated a new method to retrieve upper tropospheric humidity (UTH) from HIRS channel 12 brightness temperatures in an effort to bridge the channel s central wavelength change from $6.7\,\mu m$ to $6.5\,\mu m$ between HIRS/2 and HIRS/3 instruments. The new method retains the second-order term in the linearisation of the UTH formulation. The addition of the second-order term and the choice of the spectroscopic constant for the retrievals are shown successfully matching the UTH between HIRS/2 and HIRS/3 observations. We have added this piece of information in the introduction.

In the following we reply to other issues raised in the comments.

**2.1 Cloud influence**

*The retrieval of UTH from 6.3 μm observations is only possible when there is no clouds in the free troposphere. Brogniez and co-authors discussed at length that point in a 2006 study, where they show that clouds below ∼ 700 hPa are almost "unseen" in that channel. Here the study focuses on the cold range of the 6.3 μm temperatures. Can the authors discuss the cloud screening technique? Can't the very low brightness temperatures mentioned (230K and below) be reasonably associated to partially cloud-covered pixels? In their 2016 work, Chung et al mention the problem of the cloud detection in the HIRS serie which is done thanks to ISCCP. Therefore, my question here is: would that be possible that the discrepancy in the cold range values would be induced by the cloud screening (and thus ISCCP cloud detection), from HIRS2 to HIRS4?*

All HIRS data that we obtained from NOAA and used in this study are cloud-cleared and limb corrected. Shi and Bates (2011) write:

> The HIRS data are first processed to remove cloudy pixels for the water vapor field. The cloud clearing procedure follows the method detailed by Jackson et al. (2003). The process is accomplished using a simplified method based on the ISCCP cloud detection approach (Rossow and Garder, 1993). This approach combines spatial and temporal variations in the brightness temperature and applies thresholds to these variations to detect clouds. Limb correction is applied with a linear multivariate regression algorithm using multiple HIRS channels Jackson et al. (2003).

It is now mentioned in the revised version that cloud-clear sky is assumed in the derivations of the radiative transfer equations, it is also mentioned that our applications are perfomed with cloud-cleared data, and that for instance the detected trends refer to clear-sky. We believe that this suffices. We will not enter into a discussion about the cloud-clearing process since it is not our work.

Of course, one can never be sure that there are overlooked issues in the data, e.g. cloud influence. However, it appears that clouds are probably not the reason for the jump in the occurrence frequency of high UTH cases. In terms of brightness temperatures the jump occurs at the low end of the data, at about 230 K. Thus if clouds would explain the jump, it must be clouds at about that low temperature. Now the peak of both the HIRS 2 and HIRS 3 filter response functions is typically at a lower level, that is, an optical thickness of unity is reached below the 230 K isotherm. Assume now that an optically thick cloud is at 230 K. (Of course, thick clouds will easily be detected by any cloud clearance algorithm; we assume thick clouds here only for the development of the argument.) Then both instruments receive photons only from this layer and from further above. That is, the registred brightness temperatures are both about 230 K, which is, they are equal. This shows that a high cloud tends to bring the two brightness temperatures together (thick clouds, but thin clouds as well, only to a lesser extent). Gierens and Eleftheratos (2017) showed, however, that the two brightness temperatures are too far apart at the low end of the $T_b$ distribution, and then we conceived a correction that brings them closer together. From this argument it appears unlikely that clouds would produce the excess of high UTH cases in HIRS 3 data.

Furthermore, a potential effect of clouds is just speculation. On the contrary, it is clear that $k_\lambda$ cannot be the same for two wavelengths with quite different opacities. As we show, an appropriate choice of the optical constant leads to a satisfying comparison between UTHi from NOAA 14 and from NOAA 15. To our view it is not justified to give up this physically motivated change in favour of speculation about cloud influence.

**2.2 Historical papers**

*When reading UTH-related papers I am always stunned that people don't give credit where credit is due. I agree that B. Soden & co-authors developed the first analytic retrieval of UTH, but the retrieval of UTH from $6.3\,\mu m$ observations was first developed by J. Schmetz and O. Turpeinen in 1988 and produced operationally at EUMETSAT ever since. I deeply regret such evolution in the referencing. It is obvious that every paper cannot list all the previous work performed on a particular topic, but then when the paper goes back to the roots, then I totally believe that this has to be done.*

We now include a new paragraph in the introduction that covers the evolution of UTH retrieval from the beginning of the era of meteorological satellites until Schmetz and Turpeinen (1988), who probably invented the term UTH.

**2.3 Weighting function**

*... Brogniez et al 2009 and (maybe closer to us) Schröder et al 2014 that have redefined the weighting function by showing that the transmission-derived weighting function of SB93 was the least accurate one, that didn't consider the radiatively-driven information, as the authors underline it. But then the part 5.3 (as well as 5.4...) is, in my opinion, a little bit "reheated"... ⇒ Can the author go through the definition proposed in Schröder's work for instance and compare to their definition of the weighting function?*

The referee's comments on the use of the weighting function made us rethinking the issue and indeed we found that section 5.3 contained unneccessary considerations. But in this process we found new aspects and thus a) we replaced the title of 5.3, b) changed the text almost completely and c) removed Figures 12 and 13. The last sentence of the abstract and the last item in the enumerated list of the conclusions are deleted accordingly.

Further we want to point at a certain confusion that people seem to have with the notion weighting function (cf. Gierens et al., 2018, supplement to the paper). There are several quantities that can be weighted and the meaning of "weighting function" depends on what is weighted. Let us give some examples:

- The radiative transfer equation involves the integral

$$\int B(x)\,(d\mathcal{T}(x)/dx)\,\mathrm{d}x$$

  where $\mathcal{T}(x)$ is the transmission function. The integrand represents the contribution of each differential layer $\mathrm{d}x$ to the flux of photons at the satellite. Our function $\Phi(x;U)/(I/B_0)$ is of this kind. In the old literature we found the expression "contribution function" for it (unfortunately we forgot where).

- Often the term $d\mathcal{T}(x)/dx$ is called weighting function (e.g. Conrath, 1969; Hayden et al., 1981; Harries, 1997). Jackson and Bates (2001) call it transmission function. The generic weighting function $K(...)$ is of this type. It weighs the emission (Planck function) itself.

- Jacobians $\partial I/\partial Q(x)$ measure the sensitivity of the radiance to a radiatively important quantity $Q$ at $x$. Although they are used as weighting functions to determine weighted averages of relative humidity from profiles $r(x)$, they are not weighting functions that result directly from the equation of radiative transfer. They do neither weight the emission nor the flux of photons.

The conceptual difference between the first two and the last example should be noted. Ours and Schröders use of the notion lie on opposite sides of this border and thus there is no sense in a quantitative comparison.

**2.4  Non-specified assumptions**

*Section 2, present the 1st order retrieval, as designed by SB93 and SJW96. ⇒ Eq (1) and (2) are from the Malkmus band model, adapted to strong absorbing lines, not specified. Since this section intends to re-discuss the SB93 & SJW96 works, then all the assumptions need to be written. ⇒ It is nowhere specified that the developments by SB93 and SJW96 are adapted for tropical and subtropical standard atmospheres: in these regions the temperature profile doesn't change much and that is why the $6.3\,\mu m$ observations can be interpreted as UTH. However, the authors have applied it to northern mid-latitudes. I would like to know how the approximations, and more specifically the linearizations, translate from the tropics to the mid-latitudes. For example, the pressure at which $T0 = 240K$ varies slightly between the tropics and the mid-latitudes. Is there an impact?*

Again we agree that it might be quite interesting to list all the assumptions that were necessary to derive the retrieval, for instance the assumption that the water vapour spectral lines between 6 and $7\,\mu m$ are in the square root regime of the growth curve. However, these assumptions are already presented by the authors quoted. In our paper we mention those assumptions that were changed or made more precise; there is no need to mention assumptions that we do not touch upon.

We agree that the original retrieval method was developed for the tropics. But it is also true that Jackson and Bates (2001) introduced their update to the retrieval formula for application outside the tropics. Thus we use it.

Finally, we are not chasing the TRUTH, as should be clear from the discussion section. We seek consistency in the time series for the complete range of UTHi values. Consistency could be achieved as well with slightly different $T_0$, for instance, but whether these would result in "better" UTH is not clear at all. We think furthermore that it is not desirable to introduce different retrievals for different latitude zones, which would bear the danger of getting other kinds of inconsistencies. A method like that of Jackson and Bates (2001) that just uses another HIRS brightness temperature to adapt for changing lapse rates, is appropriate, but simple jumps in parameters are not.

**3  Reply to reviewer 3**

- Page 1: UTHi will be defined in the revised version. It is the upper-tropospheric humidity with respect to ice.

- Page 2: Thanks. Corrected.

- Page 4: Determination of the optical constant was quite tedious. The goal was to achieve a bivariate regression with a slope as close to unity as possible in the N14-N15 comparison (Figure 9). For this we selected a value, made all the calculations described in the paper, including the determination of the fit coefficients and ran the comparison over the 1004 days. Then we selected another constant and looked whether the regression slope was closer to unity. This was done in an iterative way. The slope is sensitive to the choice of the optical constant, but unfortunately, we have not recorded the intermediate results, so we cannot give a quantitative answer without doing all the work again.

- Page 8: Values of the $a\prime$ and $b\prime$ coefficients are given in the revised version: $a' = 10.236, b' = -0.036$.

- Page 20: We understand the problem. Indeed the sentence "Solving for $U$..." is misleading, since the equation cannot be solved for $U$. What is meant is the following. We compute the radiance and the corresponding brightness temperature for every integer value of $U$ from 1 to 99%, using equations 22 and 25. Thus we arrive at two columns of data, with values of $U$ in one column and values of $T_{12}$ in the other. These columns are plotted against each other in Figure 5. We will change the text after eq. 25 to clarify the issue.

- Page 25: Corrected.

[revised manuscript text omitted]